# GraphSAINT: Graph Sampling Based Inductive Learning Method

**Hanqing Zeng**[*]
University of Southern California
zengh@usc.edu

**Hongkuan Zhou**[*]
University of Southern California
hongkuaz@usc.edu

**Ajitesh Srivastava**
University of Southern California
ajiteshs@usc.edu

**Rajgopal Kannan**
US Army Research Lab
rajgopal.kannan.civ@mail.mil

**Viktor Prasanna**
University of Southern California
prasanna@usc.edu

## Abstract

Graph Convolutional Networks (GCNs) are powerful models for learning representations of attributed graphs. To scale GCNs to large graphs, state-of-the-art methods use various *layer sampling* techniques to alleviate the "neighbor explosion" problem during minibatch training. We propose GraphSAINT, a *graph sampling* based inductive learning method that improves training efficiency and accuracy in a fundamentally different way. By changing perspective, GraphSAINT constructs minibatches by sampling the training graph, rather than the nodes or edges across GCN layers. Each iteration, a complete GCN is built from the properly sampled subgraph. Thus, we ensure fixed number of well-connected nodes in all layers. We further propose normalization technique to eliminate bias, and sampling algorithms for variance reduction. Importantly, we can decouple the sampling from the forward and backward propagation, and extend GraphSAINT with many architecture variants (e.g., graph attention, jumping connection). GraphSAINT demonstrates superior performance in both accuracy and training time on five large graphs, and achieves new state-of-the-art F1 scores for PPI (0.995) and Reddit (0.970).

## 1 Introduction

Recently, representation learning on graphs has attracted much attention, since it greatly facilitates tasks such as classification and clustering (Wu et al., 2019; Cai et al., 2017). Current works on Graph Convolutional Networks (GCNs) (Hamilton et al., 2017; Chen et al., 2018b; Gao et al., 2018; Huang et al., 2018; Chen et al., 2018a) mostly focus on shallow models (2 layers) on relatively small graphs. Scaling GCNs to larger datasets and deeper layers still requires fast alternate training methods.

In a GCN, data to be gathered for one output node comes from its neighbors in the previous layer. Each of these neighbors in turn, gathers its output from the previous layer, and so on. Clearly, the deeper we back track, the more multi-hop neighbors to support the computation of the root. The number of support nodes (and thus the training time) potentially grows exponentially with the GCN depth. To mitigate such "neighbor explosion", state-of-the-art methods use various *layer sampling* techniques. The works by Hamilton et al. (2017); Ying et al. (2018a); Chen et al. (2018a) ensure that only a small number of neighbors (typically from 2 to 50) are selected by one node in the next layer. Chen et al. (2018b) and Huang et al. (2018) further propose samplers to restrict the neighbor expansion factor to 1, by ensuring a fixed sample size in all layers. While these methods significantly speed up training, they face challenges in scalability, accuracy or computation complexity.

---

[*]Equal contribution

**Present work**   We present GraphSAINT (Graph SAmpling based INductive learning meThod) to efficiently train deep GCNs. GraphSAINT is developed from a fundamentally different way of minibatch construction. Instead of building a GCN on the full training graph and then sampling across the layers, we sample the training graph first and then build a full GCN on the subgraph. Our method is thus *graph sampling* based. Naturally, GraphSAINT resolves "neighbor explosion", since every GCN of the minibatches is a small yet *complete* one. On the other hand, graph sampling based method also brings new challenges in training. Intuitively, nodes of higher influence on each other should have higher probability to form a subgraph. This enables the sampled nodes to "support" each other without going outside the minibatch. Unfortunately, such strategy results in non-identical node sampling probability, and introduces bias in the minibatch estimator. To address this issue, we develop normalization techniques so that the feature learning does not give preference to nodes more frequently sampled. To further improve training quality, we perform variance reduction analysis, and design light-weight sampling algorithms by quantifying "influence" of neighbors. Experiments on GraphSAINT using five large datasets show significant performance gain in both training accuracy and time. We also demonstrate the flexibility of GraphSAINT by integrating our minibatch method with popular GCN architectures such as JK-net (Xu et al., 2018) and GAT (Veličković et al., 2017). The resulting deep models achieve new state-of-the-art F1 scores on PPI (0.995) and Reddit (0.970).

## 2 RELATED WORK

A neural network model that extends convolution operation to the graph domain is first proposed by Bruna et al. (2013). Further, Kipf & Welling (2016); Defferrard et al. (2016) speed up graph convolution computation with localized filters based on Chebyshev expansion. They target relatively small datasets and thus the training proceeds in full batch. In order to scale GCNs to large graphs, layer sampling techniques (Hamilton et al., 2017; Chen et al., 2018b; Ying et al., 2018a; Chen et al., 2018a; Gao et al., 2018; Huang et al., 2018) have been proposed for efficient minibatch training. All of them follow the three meta steps: 1. Construct a complete GCN on the full training graph. 2. Sample nodes or edges of each layer to form minibatches. 3. Propagate forward and backward among the sampled GCN. Steps (2) and (3) proceed iteratively to update the weights via stochastic gradient descent. The layer sampling algorithm of GraphSAGE (Hamilton et al., 2017) performs uniform node sampling on the previous layer neighbors. It enforces a pre-defined budget on the sample size, so as to bound the minibatch computation complexity. Ying et al. (2018a) enhances the layer sampler of Hamilton et al. (2017) by introducing an importance score to each neighbor. The algorithm presumably leads to less information loss due to weighted aggregation. S-GCN (Chen et al., 2018a) further restricts neighborhood size by requiring only two support nodes in the previous layer. The idea is to use the historical activations in the previous layer to avoid redundant re-evaluation. FastGCN (Chen et al., 2018b) performs sampling from another perspective. Instead of tracking down the inter-layer connections, node sampling is performed independently for each layer. It applies importance sampling to reduce variance, and results in constant sample size in all layers. However, the minibatches potentially become too sparse to achieve high accuracy. Huang et al. (2018) improves FastGCN by an additional sampling neural network. It ensures high accuracy, since sampling is conditioned on the selected nodes in the next layer. Significant overhead may be incurred due to the expensive sampling algorithm and the extra sampler parameters to be learned.

Instead of sampling layers, the works of Zeng et al. (2018) and Chiang et al. (2019) build minibatches from subgraphs. Zeng et al. (2018) proposes a specific graph sampling algorithm to ensure connectivity among minibatch nodes. They further present techniques to scale such training on shared-memory multi-core platforms. More recently, ClusterGCN (Chiang et al., 2019) proposes graph clustering based minibatch training. During pre-processing, the training graph is partitioned into densely connected clusters. During training, clusters are randomly selected to form minibatches, and intra-cluster edge connections remain unchanged. Similar to GraphSAINT, the works of Zeng et al. (2018) and Chiang et al. (2019) do not sample the layers and thus "neighbor explosion" is avoided. Unlike GraphSAINT, both works are heuristic based, and do not account for bias due to the unequal probability of each node / edge appearing in a minibatch.

Another line of research focuses on improving model capacity. Applying attention on graphs, the architectures of Veličković et al. (2017); Zhang et al. (2018); Lu et al. (2019) better capture neighbor features by dynamically adjusting edge weights. Klicpera et al. (2018) combines PageRank with GCNs to enable efficient information propagation from many hops away. To develop deeper models,

"skip-connection" is borrowed from CNNs (He et al., 2015; Huang et al., 2017) into the GCN context. In particular, JK-net Xu et al. (2018) demonstrates significant accuracy improvement on GCNs with more than two layers. Note, however, that JK-net (Xu et al., 2018) follows the same sampling strategy as GraphSAGE (Hamilton et al., 2017). Thus, its training cost is high due to neighbor explosion. In addition, high order graph convolutional layers (Zhou, 2017; Lee et al., 2018; Abu-El-Haija et al., 2019) also help propagate long-distance features. With the numerous architectural variants developed, the question of how to train them efficiently via minibatches still remains to be answered.

## 3 PROPOSED METHOD: GraphSAINT

Graph sampling based method is motivated by the challenges in scalability (in terms of model depth and graph size). We analyze the bias (Section 3.2) and variance (Section 3.3) introduced by graph sampling, and thus, propose feasible sampling algorithms (Section 3.4). We show the applicability of GraphSAINT to other architectures, both conceptually (Section 4) and experimentally (Section 5.2).

In the following, we define the problem of interest and the corresponding notations. A GCN learns representation of an un-directed, attributed graph $\mathcal{G}\left(\mathcal{V}, \mathcal{E}\right)$, where each node $v \in \mathcal{V}$ has a length-$f$ attribute $\boldsymbol{x}_v$. Let $\boldsymbol{A}$ be the adjacency matrix and $\widetilde{\boldsymbol{A}}$ be the normalized one (i.e., $\widetilde{\boldsymbol{A}} = \boldsymbol{D}^{-1}\boldsymbol{A}$, and $\boldsymbol{D}$ is the diagonal degree matrix). Let the dimension of layer-$\ell$ input activation be $f^{(\ell)}$. The activation of node $v$ is $\boldsymbol{x}_v^{(\ell)} \in \mathbb{R}^{f^{(\ell)}}$, and the weight matrix is $\boldsymbol{W}^{(\ell)} \in \mathbb{R}^{f^{(\ell)} \times f^{(\ell+1)}}$. Note that $\boldsymbol{x}_v = \boldsymbol{x}_v^{(1)}$. Propagation rule of a layer is defined as follows:

$$\boldsymbol{x}_v^{(\ell+1)} = \sigma \left( \sum_{u \in \mathcal{V}} \widetilde{\boldsymbol{A}}_{v,u} \left( \boldsymbol{W}^{(\ell)} \right)^\mathsf{T} \boldsymbol{x}_u^{(\ell)} \right) \tag{1}$$

where $\widetilde{\boldsymbol{A}}_{v,u}$ is a scalar, taking an element of $\widetilde{\boldsymbol{A}}$. And $\sigma\left(\cdot\right)$ is the activation function (e.g., ReLU).

We use subscript "s" to denote parameterd of the sampled graph (e.g., $\mathcal{G}_s, \mathcal{V}_s, \mathcal{E}_s$).

GCNs can be applied under inductive and transductive settings. While GraphSAINT is applicable to both, in this paper, we focus on inductive learning. It has been shown that inductive learning is especially challenging (Hamilton et al., 2017) — during training, neither attributes nor connections of the test nodes are present. Thus, an inductive model has to generalize to completely unseen graphs.

### 3.1 MINIBATCH BY GRAPH SAMPLING

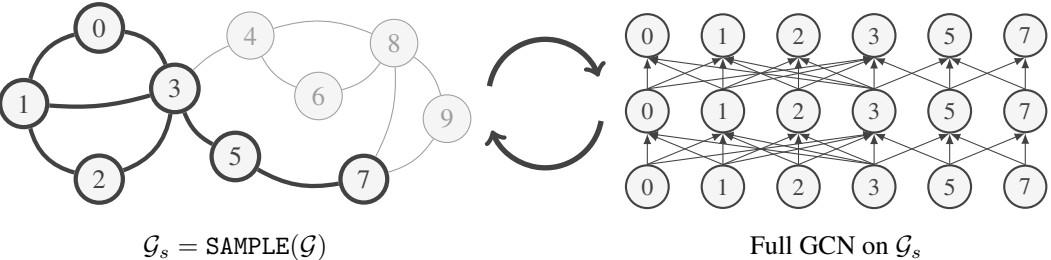

$$\mathcal{G}_s = \texttt{SAMPLE}(\mathcal{G}) \qquad\qquad\qquad \text{Full GCN on } \mathcal{G}_s$$

Figure 1: GraphSAINT training algorithm

GraphSAINT follows the design philosophy of directly sampling the training graph $\mathcal{G}$, rather than the corresponding GCN. Our goals are to 1. extract appropriately connected subgraphs so that little information is lost when propagating within the subgraphs, and 2. combine information of many subgraphs together so that the training process overall learns good representation of the full graph.

Figure 1 and Algorithm 1 illustrate the training algorithm. Before training starts, we perform light-weight pre-processing on $\mathcal{G}$ with the given sampler SAMPLE. The pre-processing estimates the probability of a node $v \in \mathcal{V}$ and an edge $e \in \mathcal{E}$ being sampled by SAMPLE. Such probability is later used to normalize the subgraph neighbor aggregation and the minibatch loss (Section 3.2). Afterwards,

---

**Algorithm 1** `GraphSAINT` training algorithm

---

**Input:** Training graph $\mathcal{G}(\mathcal{V}, \mathcal{E}, \boldsymbol{X})$; Labels $\overline{\boldsymbol{Y}}$; Sampler `SAMPLE`;
**Output:** GCN model with trained weights
 1: Pre-processing: Setup `SAMPLE` parameters; Compute normalization coefficients $\alpha$, $\lambda$.
 2: **for** each minibatch **do**
 3:     $\mathcal{G}_s(\mathcal{V}_s, \mathcal{E}_s) \leftarrow$ Sampled sub-graph of $\mathcal{G}$ according to `SAMPLE`
 4:     GCN construction on $\mathcal{G}_s$.
 5:     $\{\boldsymbol{y}_v \mid v \in \mathcal{V}_s\} \leftarrow$ Forward propagation of $\{\boldsymbol{x}_v \mid v \in \mathcal{V}_s\}$, normalized by $\alpha$
 6:     Backward propagation from $\lambda$-normalized loss $L(\boldsymbol{y}_v, \overline{\boldsymbol{y}}_v)$. Update weights.
 7: **end for**

---

training proceeds by iterative weight updates via SGD. Each iteration starts with an independently sampled $\mathcal{G}_s$ (where $|\mathcal{V}_s| \ll |\mathcal{V}|$). We then build a full GCN on $\mathcal{G}_s$ to generate embedding and calculate loss for every $v \in \mathcal{V}_s$. In Algorithm 1, node representation is learned by performing node classification in the supervised setting, and each training node $v$ comes with a ground truth label $\overline{\boldsymbol{y}}_v$.

Intuitively, there are two requirements for `SAMPLE`: 1. Nodes having high influence on each other should be sampled in the same subgraph. 2. Each edge should have non-negligible probability to be sampled. For requirement (1), an ideal `SAMPLE` would consider the joint information from node connections as well as attributes. However, the resulting algorithm may have high complexity as it would need to infer the relationships between features. For simplicity, we define "influence" from the graph connectivity perspective and design topology based samplers. Requirement (2) leads to better generalization since it enables the neural net to explore the full feature and label space.

### 3.2 Normalization

A sampler that preserves connectivity characteristic of $\mathcal{G}$ will almost inevitably introduce bias into minibatch estimation. In the following, we present normalization techniques to eliminate biases.

Analysis of the complete multi-layer GCN is difficult due to non-linear activations. Thus, we analyze the embedding of each layer independently. This is similar to the treatment of layers independently by prior work (Chen et al., 2018b; Huang et al., 2018). Consider a layer-$(\ell + 1)$ node $v$ and a layer-$\ell$ node $u$. If $v$ is sampled (i.e., $v \in \mathcal{V}_s$), we can compute the aggregated feature of $v$ as:

$$\zeta_v^{(\ell+1)} = \sum_{u \in \mathcal{V}} \frac{\widetilde{\boldsymbol{A}}_{v,u}}{\alpha_{u,v}} \left(\boldsymbol{W}^{(\ell)}\right)^{\mathsf{T}} \boldsymbol{x}_u^{(\ell)} \mathbb{1}_{u|v} = \sum_{u \in \mathcal{V}} \frac{\widetilde{\boldsymbol{A}}_{v,u}}{\alpha_{u,v}} \tilde{\boldsymbol{x}}_u^{(\ell)} \mathbb{1}_{u|v}, \tag{2}$$

where $\tilde{\boldsymbol{x}}_u^{(\ell)} = \left(\boldsymbol{W}^{(\ell)}\right)^{\mathsf{T}} \boldsymbol{x}_u^{(\ell)}$, and $\mathbb{1}_{u|v} \in \{0, 1\}$ is the indicator function given $v$ is in the subgraph (i.e., $\mathbb{1}_{u|v} = 0$ if $v \in \mathcal{V}_s \wedge (u,v) \notin \mathcal{E}_s$; $\mathbb{1}_{u|v} = 1$ if $(u,v) \in \mathcal{E}_s$; $\mathbb{1}_{u|v}$ not defined if $v \notin \mathcal{V}_s$). We refer to the constant $\alpha_{u,v}$ as *aggregator normalization*. Define $p_{u,v} = p_{v,u}$ as the probability of an edge $(u,v) \in \mathcal{E}$ being sampled in a subgraph, and $p_v$ as the probability of a node $v \in \mathcal{V}$ being sampled.

**Proposition 3.1.** $\zeta_v^{(\ell+1)}$ *is an unbiased estimator of the aggregation of $v$ in the full* $(\ell+1)^{th}$ *GCN layer, if* $\alpha_{u,v} = \frac{p_{u,v}}{p_v}$. *i.e.,* $\mathbb{E}\left(\zeta_v^{(\ell+1)}\right) = \sum_{u \in \mathcal{V}} \widetilde{\boldsymbol{A}}_{v,u} \tilde{\boldsymbol{x}}_u^{(\ell)}$.

Assuming that each layer independently learns an embedding, we use Proposition 3.1 to normalize feature propagation of each layer of the GCN built by `GraphSAINT`. Further, let $L_v$ be the loss on $v$ in the output layer. The minibatch loss is calculated as $L_{\text{batch}} = \sum_{v \in \mathcal{G}_s} L_v / \lambda_v$, where $\lambda_v$ is a constant that we term *loss normalization*. We set $\lambda_v = |\mathcal{V}| \cdot p_v$ so that:

$$\mathbb{E}(L_{\text{batch}}) = \frac{1}{|\mathbb{G}|} \sum_{\mathcal{G}_s \in \mathbb{G}} \sum_{v \in \mathcal{V}_s} \frac{L_v}{\lambda_v} = \frac{1}{|\mathcal{V}|} \sum_{v \in \mathcal{V}} L_v. \tag{3}$$

Feature propagation within subgraphs thus requires normalization factors $\alpha$ and $\lambda$, which are computed based on the edge and node probability $p_{u,v}$, $p_v$. In the case of random node or random edge samplers, $p_{u,v}$ and $p_v$ can be derived analytically. For other samplers in general, closed form expression is hard to obtain. Thus, we perform pre-processing for estimation. Before training starts,

we run the sampler repeatedly to obtain a set of $N$ subgraphs $\mathbb{G}$. We setup a counter $C_v$ and $C_{u,v}$ for each $v \in \mathcal{V}$ and $(u, v) \in \mathcal{E}$, to count the number of times the node or edge appears in the subgraphs of $\mathbb{G}$. Then we set $\alpha_{u,v} = \frac{C_{u,v}}{C_v} = \frac{C_{v,u}}{C_v}$ and $\lambda_v = \frac{C_v}{N}$. The subgraphs $\mathcal{G}_s \in \mathbb{G}$ can all be reused as minibatches during training. Thus, the overhead of pre-processing is small (see Appendix D.2).

## 3.3 VARIANCE

We derive samplers for variance reduction. Let $e$ be the edge connecting $u, v$, and $\boldsymbol{b}_e^{(\ell)} = \widetilde{\boldsymbol{A}}_{v,u}\tilde{\boldsymbol{x}}_u^{(\ell-1)} + \widetilde{\boldsymbol{A}}_{u,v}\tilde{\boldsymbol{x}}_v^{(\ell-1)}$. It is desirable that variance of all estimators $\zeta_v^{(\ell)}$ is small. With this objective, we define:

$$\zeta = \sum_\ell \sum_{v \in \mathcal{G}_s} \frac{\zeta_v^{(\ell)}}{p_v} = \sum_\ell \sum_{v,u} \frac{\widetilde{\boldsymbol{A}}_{v,u}}{p_v \alpha_{u,v}} \tilde{\boldsymbol{x}}_u^{(\ell)} \mathbb{1}_v \mathbb{1}_{u|v} = \sum_\ell \sum_e \frac{\boldsymbol{b}_e^{(\ell)}}{p_e} \mathbb{1}_e^{(\ell)}. \tag{4}$$

where $\mathbb{1}_e = 1$ if $e \in \mathcal{E}_s$; $\mathbb{1}_e = 0$ if $e \notin \mathcal{E}_s$. And $\mathbb{1}_v = 1$ if $v \in \mathcal{V}_s$; $\mathbb{1}_v = 0$ if $v \notin \mathcal{V}_s$. The factor $p_u$ in the first equality is present so that $\zeta$ is an unbiased estimator of the sum of all node aggregations at all layers: $\mathbb{E}(\zeta) = \sum_\ell \sum_{v \in \mathcal{V}} \mathbb{E}\left(\zeta_v^{(\ell)}\right)$. Note that $\mathbb{1}_e^{(\ell)} = \mathbb{1}_e, \forall \ell$, since once an edge is present in the sampled graph, it is present in all layers of our GCN.

We define the optimal edge sampler to minimize variance for every dimension of $\zeta$. We restrict ourselves to independent edge sampling. For each $e \in \mathcal{E}$, we make independent decision on whether it should be in $\mathcal{G}_s$ or not. The probability of including $e$ is $p_e$. We further constrain $\sum p_e = m$, so that the expected number of sampled edges equals to $m$. The budget $m$ is a given sampling parameter.

**Theorem 3.2.** *Under independent edge sampling with budget $m$, the optimal edge probabilities to minimize the sum of variance of each $\zeta$'s dimension is given by: $p_e = \frac{m}{\sum_{e'}\left\|\sum_\ell \boldsymbol{b}_{e'}^{(\ell)}\right\|}\left\|\sum_\ell \boldsymbol{b}_e^{(\ell)}\right\|$.*

To prove Theorem 3.2, we make use of the independence among graph edges, and the dependence among layer edges to obtain the covariance of $\mathbb{1}_e^{(\ell)}$. Then using the fact that sum of $p_e$ is a constant, we use the Cauchy-Schwarz inequality to derive the optimal $p_e$. Details are in Appendix A.

Note that calculating $\boldsymbol{b}_e^{(\ell)}$ requires computing $\tilde{\boldsymbol{x}}_v^{(\ell-1)}$, which increases the complexity of sampling. As a reasonable simplification, we ignore $\tilde{\boldsymbol{x}}_v^{(\ell)}$ to make the edge probability dependent on the graph topology only. Therefore, we choose $p_e \propto \widetilde{\boldsymbol{A}}_{v,u} + \widetilde{\boldsymbol{A}}_{u,v} = \frac{1}{\deg(u)} + \frac{1}{\deg(v)}$.

The derived optimal edge sampler agrees with the intuition in Section 3.1. If two nodes $u, v$ are connected and they have few neighbors, then $u$ and $v$ are likely to be influential to each other. In this case, the edge probability $p_{u,v} = p_{v,u}$ should be high. The above analysis on edge samplers also inspires us to design other samplers, which are presented in Section 3.4.

**Remark** We can also apply the above edge sampler to perform layer sampling. Under the independent layer sampling assumption of Chen et al. (2018b), one would sample a connection $\left(u^{(\ell)}, v^{(\ell+1)}\right)$ with probability $p_{u,v}^{(\ell)} \propto \frac{1}{\deg(u)} + \frac{1}{\deg(v)}$. For simplicity, assume a uniform degree graph (of degree $d$). Then $p_e^{(\ell)} = p$. For an already sampled $u^{(\ell)}$ to connect to layer $\ell + 1$, at least one of its edges has to be selected by the layer $\ell + 1$ sampler. Clearly, the probability of an input layer node to "survive" the $L$ number of independent sampling process is $\left(1 - (1-p)^d\right)^{L-1}$. Such layer sampler potentially returns an overly sparse minibatch for $L > 1$. On the other hand, connectivity within a minibatch of GraphSAINT never drops with GCN depth. If an edge is present in layer $\ell$, it is present in all layers.

## 3.4 SAMPLERS

Based on the above variance analysis, we present several light-weight and efficient samplers that GraphSAINT has integrated. Detailed sampling algorithms are listed in Appendix B.

**Random node sampler** We sample $|\mathcal{V}_s|$ nodes from $\mathcal{V}$ randomly, according to a node probability distribution $P(u) \propto \left\|\widetilde{\boldsymbol{A}}_{:,u}\right\|^2$. This sampler is inspired by the layer sampler of Chen et al. (2018b).

**Random edge sampler**    We perform edge sampling as described in Section 3.3.

**Random walk based samplers**    Another way to analyze graph sampling based multi-layer GCN is to ignore activations. In such case, $L$ layers can be represented as a single layer with edge weights given by $\boldsymbol{B} = \widetilde{\boldsymbol{A}}^L$. Following a similar approach as Section 3.3, if it were possible to pick pairs of nodes (whether or not they are directly connected in the original $\widetilde{\boldsymbol{A}}$) independently, then we would set $p_{u,v} \propto \boldsymbol{B}_{u,v} + \boldsymbol{B}_{v,u}$, where $\boldsymbol{B}_{u,v}$ can be interpreted as the probability of a random walk to start at $u$ and end at $v$ in $L$ hops (and $\boldsymbol{B}_{v,u}$ vice-versa). Even though it is not possible to sample a subgraph where such pairs of nodes are independently selected, we still consider a random walk sampler with walk length $L$ as a good candidate for $L$-layer GCNs. There are numerous random walk based samplers proposed in the literature (Ribeiro & Towsley, 2010; Leskovec & Faloutsos, 2006; Hu & Lau, 2013; Li et al., 2015). In the experiments, we implement a regular random walk sampler (with $r$ root nodes selected uniformly at random and each walker goes $h$ hops), and also a multi-dimensional random walk sampler defined in Ribeiro & Towsley (2010).

For all the above samplers, we return the subgraph induced from the sampled nodes. The induction step adds more connections into the subgraph, and empirically helps improve convergence.

## 4    Discussion

**Extensions**    `GraphSAINT` admits two orthogonal extensions. First, `GraphSAINT` can seamlessly integrate other graph samplers. Second, the idea of training by graph sampling is applicable to many GCN architecture variants: 1. **Jumping knowledge** (Xu et al., 2018): since our GCNs constructed during training are complete, applying skip connections to `GraphSAINT` is straightforward. On the other hand, for some layer sampling methods (Chen et al., 2018b; Huang et al., 2018), extra modification to their samplers is required, since the jumping knowledge architecture requires layer-$\ell$ samples to be a subset of layer-$(\ell - 1)$ samples[*]. 2. **Attention** (Veličković et al., 2017; Fey, 2019; Zhang et al., 2018): while explicit variance reduction is hard due to the dynamically updated attention values, it is reasonable to apply attention within the subgraphs which are considered as representatives of the full graph. Our loss and aggregator normalizations are also applicable[†]. 3. **Others**: To support high order layers (Zhou, 2017; Lee et al., 2018; Abu-El-Haija et al., 2019) or even more complicated networks for the task of graph classification (Ying et al., 2018b), we replace the full adjacency matrix $\boldsymbol{A}$ with the (normalized) one for the subgraph $\boldsymbol{A}_s$ to perform layer propagation.

**Comparison**    `GraphSAINT` enjoys: 1. high scalability and efficiency, 2. high accuracy, and 3. low training complexity. Point (1) is due to the significantly reduced neighborhood size compared with Hamilton et al. (2017); Ying et al. (2018a); Chen et al. (2018a). Point (2) is due to the better inter-layer connectivity compared with Chen et al. (2018b), and unbiased minibatch estimator compared with Chiang et al. (2019). Point (3) is due to the simple and trivially parallelizable pre-processing compared with the sampling of Huang et al. (2018) and clustering of Chiang et al. (2019).

## 5    Experiments

**Setup**    Experiments are under the inductive, supervised learning setting. We evaluate `GraphSAINT` on the following tasks: 1. classifying protein functions based on the interactions of human tissue proteins (PPI), 2. categorizing types of images based on the descriptions and common properties of online images (Flickr), 3. predicting communities of online posts based on user comments (Reddit), 4. categorizing types of businesses based on customer reviewers and friendship (Yelp), and 5. classifying product categories based on buyer reviewers and interactions (Amazon). For PPI, we use the small version for the two layer convergence comparison (Table 2 and Figure 2), since Hamilton et al. (2017) and Chen et al. (2018a) report accuracy for this version in their original papers. We use the large version for additional comparison with Chiang et al. (2019) to be consistent with its reported accuracy. All datasets follow "fixed-partition" splits. Appendix C.2 includes further details.

---

[*]The skip-connection design proposed by Huang et al. (2018) does not have such "subset" requirement, and thus is compatible with both graph sampling and layer sampling based methods.

[†]When applying `GraphSAINT` to GAT (Veličković et al., 2017), we remove the softmax step which normalizes attention values within the same neighborhood, as suggested by Huang et al. (2018). See Appendix C.3.

Table 1: Dataset statistics ("m" stands for **m**ulti-class classification, and "s" for **s**ingle-class.)

| Dataset | Nodes | Edges | Degree | Feature | Classes | Train / Val / Test |
|---|---|---|---|---|---|---|
| PPI | 14,755 | 225,270 | 15 | 50 | 121 (m) | 0.66 / 0.12 / 0.22 |
| Flickr | 89,250 | 899,756 | 10 | 500 | 7 (s) | 0.50 / 0.25 / 0.25 |
| Reddit | 232,965 | 11,606,919 | 50 | 602 | 41 (s) | 0.66 / 0.10 / 0.24 |
| Yelp | 716,847 | 6,977,410 | 10 | 300 | 100 (m) | 0.75 / 0.10 / 0.15 |
| Amazon | 1,598,960 | 132,169,734 | 83 | 200 | 107 (m) | 0.85 / 0.05 / 0.10 |
| PPI (large version) | 56,944 | 818,716 | 14 | 50 | 121 (m) | 0.79 / 0.11 / 0.10 |

We open source `GraphSAINT`[‡]. We compare with six baselines: 1. vanilla GCN (Kipf & Welling, 2016), 2. GraphSAGE (Hamilton et al., 2017), 3. FastGCN (Chen et al., 2018b), 4. S-GCN (Chen et al., 2018a), 5. AS-GCN (Huang et al., 2018), and 6. ClusterGCN (Chiang et al., 2019). All baselines are executed with their officially released code (see Appendix C.3 for downloadable URLs and commit numbers). Baselines and `GraphSAINT` are all implemented in Tensorflow with Python3. We run experiments on a NVIDIA Tesla P100 GPU (see Appendix C.1 for hardware specification).

## 5.1 COMPARISON WITH STATE-OF-THE-ART

Table 2 and Figure 2 show the accuracy and convergence comparison of various methods. All results correspond to two-layer GCN models (for GraphSAGE, we use its mean aggregator). For a given dataset, we keep hidden dimension the same across all methods. We describe the detailed architecture and hyperparameter search procedure in Appendix C.3. The mean and confidence interval of the accuracy values in Table 2 are measured by three runs under the same hyperparameters. The training time of Figure 2 excludes the time for data loading, pre-processing, validation set evaluation and model saving. Our pre-processing incurs little overhead in training time. See Appendix D.2 for cost of graph sampling. For `GraphSAINT`, we implement the graph samplers described in Section 3.4. In Table 2, "Node" stands for random node sampler; "Edge" stands for random edge sampler; "RW" stands for random walk sampler; "MRW" stands for multi-dimensional random walk sampler.

Table 2: Comparison of test set F1-micro score with state-of-the-art methods

| Method | PPI | Flickr | Reddit | Yelp | Amazon |
|---|---|---|---|---|---|
| GCN | 0.515±0.006 | 0.492±0.003 | 0.933±0.000 | 0.378±0.001 | 0.281±0.005 |
| GraphSAGE | 0.637±0.006 | 0.501±0.013 | 0.953±0.001 | 0.634±0.006 | 0.758±0.002 |
| FastGCN | 0.513±0.032 | 0.504±0.001 | 0.924±0.001 | 0.265±0.053 | 0.174±0.021 |
| S-GCN | 0.963±0.010 | 0.482±0.003 | 0.964±0.001 | 0.640±0.002 | —[‡] |
| AS-GCN | 0.687±0.012 | 0.504±0.002 | 0.958±0.001 | —[‡] | —[‡] |
| ClusterGCN | 0.875±0.004 | 0.481±0.005 | 0.954±0.001 | 0.609±0.005 | 0.759±0.008 |
| `GraphSAINT-Node` | 0.960±0.001 | 0.507±0.001 | 0.962±0.001 | 0.641±0.000 | 0.782±0.004 |
| `GraphSAINT-Edge` | **0.981**±0.007 | 0.510±0.002 | **0.966**±0.001 | **0.653**±0.003 | 0.807±0.001 |
| `GraphSAINT-RW` | **0.981**±0.004 | **0.511**±0.001 | **0.966**±0.001 | **0.653**±0.003 | **0.815**±0.001 |
| `GraphSAINT-MRW` | 0.980±0.006 | 0.510±0.001 | 0.964±0.000 | 0.652±0.001 | 0.809±0.001 |

Table 3: Additional comparison with ClusterGCN (test set F1-micro score)

| | PPI (large version) | | Reddit | |
|---|---|---|---|---|
| | $2 \times 512$ | $5 \times 2048$ | $2 \times 128$ | $4 \times 128$ |
| ClusterGCN | 0.903±0.002 | 0.994±0.000 | 0.954±0.001 | 0.966±0.001 |
| `GraphSAINT` | **0.941**±0.003 | **0.995**±0.000 | **0.966**±0.001 | **0.970**±0.001 |

---

[‡]Open sourced code: `https://github.com/GraphSAINT/GraphSAINT`
[‡]The codes throw runtime error on the large datasets (Yelp or Amazon).

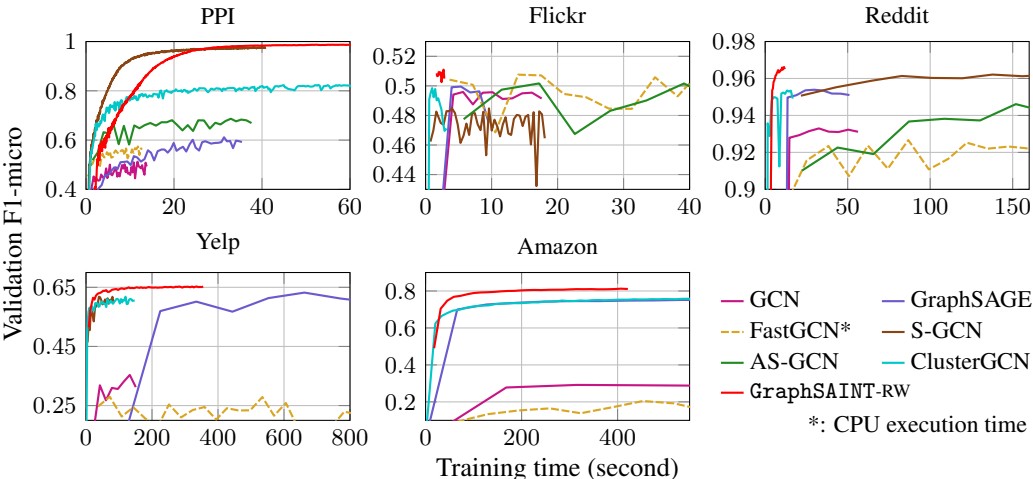

Figure 2: Convergence curves of 2-layer models on `GraphSAINT` and baselines

Clearly, with appropriate graph samplers, `GraphSAINT` achieves significantly higher accuracy on all datasets. For `GraphSAINT`-Node, we use the same node probability as FastGCN. Thus, the accuracy improvement is mainly due to the switching from layer sampling to graph sampling (see "Remark" in Section 3.3). Compared with AS-GCN, `GraphSAINT` is significantly faster. The sampler of AS-GCN is expensive to execute, making its overall training time even longer than vanilla GCN. We provide detailed computation complexity analysis on the sampler in Appendix D.2. For S-GCN on Reddit, it achieves similar accuracy as `GraphSAINT`, at the cost of over $9\times$ longer training time. The released code of FastGCN only supports CPU execution, so its convergence curve is dashed.

Table 3 presents additional comparison with ClusterGCN. We use $L \times f$ to specify the architecture, where $L$ and $f$ denote GCN depth and hidden dimension, respectively. The four architectures are the ones used in the original paper (Chiang et al., 2019). Again, `GraphSAINT` achieves significant accuracy improvement. To train models with $L > 2$ often requires additional architectural tweaks. ClusterGCN uses its diagonal enhancement technique for the 5-layer PPI and 4-layer Reddit models. `GraphSAINT` uses jumping knowledge connection (Xu et al., 2018) for 4-layer Reddit.

**Evaluation on graph samplers**   From Table 2, random edge and random walk based samplers achieve higher accuracy than the random node sampler. Figure 3 presents sensitivity analysis on parameters of "RW". We use the same hyperparameters (except the sampling parameters) and network architecture as those of the "RW" entries in Table 2. We fix the length of each walker to 2 (i.e., GCN depth), and vary the number of roots $r$ from 250 to 2250. For PPI, increasing $r$ from 250 to 750 significantly improves accuracy. Overall, for all datasets, accuracy stabilizes beyond $r = 750$.

## 5.2   `GraphSAINT` on Architecture Variants and Deep Models

In Figure 4, we train a 2-layer and a 4-layer model of GAT (Veličković et al., 2017) and JK-net (Xu et al., 2018), by using minibatches of GraphSAGE and `GraphSAINT` respectively. On the two 4-layer architectures, `GraphSAINT` achieves two orders of magnitude speedup than GraphSAGE, indicating much better scalability on deep models. From accuracy perspective, 4-layer GAT-SAGE and JK-SAGE do not outperform the corresponding 2-layer versions, potentially due to the smoothening effect caused by the massive neighborhood size. On the other hand, with minibatches returned by our edge sampler, increasing model depth of JK-`SAINT` leads to noticeable accuracy improvement (from 0.966 of 2-layer to 0.970 of 4-layer). Appendix D.1 contains additional scalability results.

## 6   Conclusion

We have presented `GraphSAINT`, a graph sampling based training method for deep GCNs on large graphs. We have analyzed bias and variance of the minibatches defined on subgraphs, and proposed

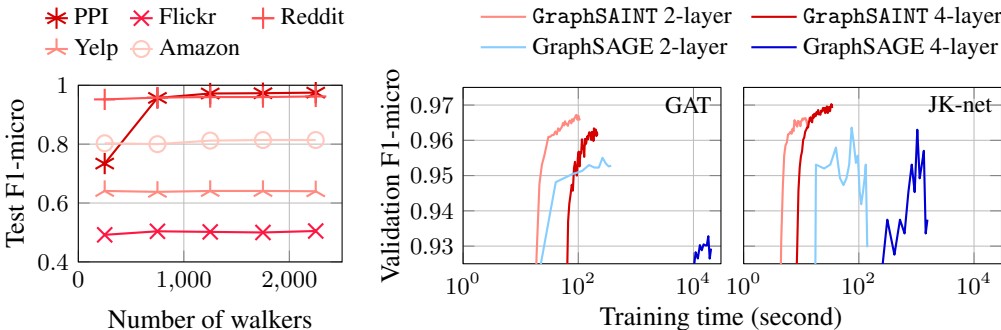

Figure 3: Sensitivity analysis          Figure 4: `GraphSAINT` with JK-net and GAT (Reddit)

normalization techniques and sampling algorithms to improve training quality. We have conducted extensive experiments to demonstrate the advantage of `GraphSAINT` in accuracy and training time.

An interesting future direction is to develop distributed training algorithms using graph sampling based minibatches. After partitioning the training graph in distributed memory, sampling can be performed independently on each processor. Afterwards, training on the self-supportive subgraphs can significantly reduce the system-level communication cost. To ensure the overall convergence quality, data shuffling strategy for the graph nodes and edges can be developed together with each specific graph sampler. Another direction is to perform algorithm-system co-optimization to accelerate the training of `GraphSAINT` on heterogeneous computing platforms (Zeng et al., 2018; Zeng & Prasanna, 2019). The resolution of "neighbor explosion" by `GraphSAINT` not only reduces the training computation complexity, but also improves hardware utilization by significantly less data traffic to the slow memory. In addition, task-level parallelization is easy since the light-weight graph sampling is completely decoupled from the GCN layer propagation.

ACKNOWLEDGEMENT

This material is based on work supported by the Defense Advanced Research Projects Agency (DARPA) under Contract Number FA8750-17-C-0086 and National Science Foundation (NSF) under Contract Numbers CCF-1919289 and OAC-1911229. Any opinions, findings and conclusions or recommendations expressed in this material are those of the authors and do not necessarily reflect the views of DARPA or NSF.

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

## A  PROOFS

*Proof of Proposition 3.1.* Under the condition that $v$ is sampled in a subgraph:

$$
\begin{aligned}
\mathbb{E}\left(\zeta_v^{(\ell+1)}\right) =& \mathbb{E}\left(\sum_{u \in \mathcal{V}} \frac{\widetilde{\boldsymbol{A}}_{v,u}}{\alpha_{u,v}} \tilde{\boldsymbol{x}}_u^{(\ell)} \mathbb{1}_{u|v}\right) \\
=& \sum_{u \in \mathcal{V}} \frac{\widetilde{\boldsymbol{A}}_{v,u}}{\alpha_{u,v}} \tilde{\boldsymbol{x}}_u^{(\ell)} \mathbb{E}\left(\mathbb{1}_{u|v}\right) \\
=& \sum_{u \in \mathcal{V}} \frac{\widetilde{\boldsymbol{A}}_{v,u}}{\alpha_{u,v}} \tilde{\boldsymbol{x}}_u^{(\ell)} P\left((u,v) \text{ sampled}|v \text{ sampled}\right) \\
=& \sum_{u \in \mathcal{V}} \frac{\widetilde{\boldsymbol{A}}_{v,u}}{\alpha_{u,v}} \tilde{\boldsymbol{x}}_u^{(\ell)} \frac{P\left((u,v) \text{ sampled}\right)}{P\left(v \text{ sampled}\right)} \\
=& \sum_{u \in \mathcal{V}} \frac{\widetilde{\boldsymbol{A}}_{v,u}}{\alpha_{u,v}} \tilde{\boldsymbol{x}}_u^{(\ell)} \frac{p_{u,v}}{p_v}
\end{aligned}
\tag{5}
$$

where the second equality is due to linearity of expectation, and the third equality (conditional edge probability) is due to the initial condition that $v$ is sampled in a subgraph.

It directly follows that, when $\alpha_{u,v} = \frac{p_{u,v}}{p_v}$,

$$
\mathbb{E}\left(\zeta_v^{(\ell+1)}\right) = \sum_{u \in \mathcal{V}} \widetilde{\boldsymbol{A}}_{v,u} \tilde{\boldsymbol{x}}_u^{(\ell)}
$$

$\square$

*Proof of Theorem 3.2.* Below, we use $\text{Cov}\left(\cdot\right)$ to denote covariance and $\text{Var}\left(\cdot\right)$ to denote variance. For independent edge sampling as defined in Section 3.3, $\text{Cov}\left(\mathbb{1}_{e_1}^{(\ell_1)}, \mathbb{1}_{e_2}^{(\ell_2)}\right) = 0, \forall e_1 \neq e_2$. And for a full GCN on the subgraph, $\text{Cov}\left(\mathbb{1}_e^{(\ell_1)}, \mathbb{1}_e^{(\ell_2)}\right) = p_e - p_e^2$. To start the proof, we first assume that the $\boldsymbol{b}_e^{(\ell)}$ is one dimensional (i.e., a scalar) and denote it by $b_e^{(\ell)}$. Now,

$$
\begin{aligned}
\text{Var}\left(\zeta\right) &= \sum_{e,\ell} \left(\frac{b_e^{(\ell)}}{p_e}\right)^2 \text{Var}\left(\mathbb{1}_e^{(\ell)}\right) + 2 \sum_{e,\ell_1<\ell_2} \frac{b_e^{(\ell_1)} b_e^{(\ell_2)}}{p_e^2} \text{Cov}\left(\mathbb{1}_e^{(\ell_1)}, \mathbb{1}_e^{(\ell_2)}\right) \\
&= \sum_{e,\ell} \frac{\left(b_e^{(\ell)}\right)^2}{p_e} - \sum_{e,\ell} \left(b_e^{(\ell)}\right)^2 + 2 \sum_{e,\ell_1<\ell_2} \frac{b_e^{(\ell_1)} b_e^{(\ell_2)}}{p_e^2} \left(p_e - p_e^2\right) \\
&= \sum_e \frac{\left(\sum_\ell b_e^{(\ell)}\right)^2}{p_e} - \sum_e \left(\sum_\ell b_e^{(\ell)}\right)^2
\end{aligned}
\tag{6}
$$

Let a given constant $m = \sum_e p_e$ be the expected number of sampled edges. By Cauchy-Schwarz inequality: $\sum_e \frac{\left(\sum_\ell b_e^{(\ell)}\right)^2}{p_e} m = \sum_e \left(\frac{\sum_\ell b_e^{(\ell)}}{\sqrt{p_e}}\right)^2 \sum_e \left(\sqrt{p_e}\right)^2 \geq \left(\sum_{e,\ell} b_e^{(\ell)}\right)^2$. The equality is achieved when $\left|\frac{\sum_\ell b_e^{(\ell)}}{\sqrt{p_e}}\right| \propto \sqrt{p_e}$. i.e., variance is minimized when $p_e \propto \left|\sum_\ell b_e^{(\ell)}\right|$.

It directly follows that:

$$
p_e = \frac{m}{\sum_{e'} \left|\sum_\ell b_{e'}^{(\ell)}\right|} \left|\sum_\ell b_e^{(\ell)}\right|
$$

For the multi-dimensional case of $\boldsymbol{b}_e^{(\ell)}$, following similar steps as above, it is easy to show that the optimal edge probability to minimize $\sum_i \text{Var}\left(\zeta_i\right)$ (where $i$ is the index for $\zeta$'s dimensions) is:

$$
p_e = \frac{m}{\sum_{e'} \left\|\sum_\ell \boldsymbol{b}_{e'}^{(\ell)}\right\|} \left\|\sum_\ell \boldsymbol{b}_e^{(\ell)}\right\|
$$

$\square$

## B  SAMPLING ALGORITHM

Algorithm 2 lists the four graph samplers we have integrated into `GraphSAINT`. The naming of the samplers follows that of Table 2. Note that the sampling parameters $n$ and $m$ specify a budget rather than the actual number of nodes and edges in the subgraph $\mathcal{G}_s$. Since certain nodes or edges in the training graph $\mathcal{G}$ may be repeatedly sampled under a single invocation of the sampler, we often have $|\mathcal{V}_s| < n$ for node and MRW samplers, $|\mathcal{V}_s| < 2m$ for edge sampler, and $|\mathcal{V}_s| < r \cdot h$ for RW sampler.

Also note that the edge sampler presented in Algorithm 2 is an approximate version of the independent edge sampler defined in Section 3.4. Complexity (excluding the subgraph induction step) of the original version in Section 3.4 is $\mathcal{O}\left(|\mathcal{E}|\right)$, while complexity of the approximate one is $\mathcal{O}\left(m\right)$. When $m \ll |\mathcal{E}|$, the approximate version leads to identical accuracy as the original one, for a given $m$.

## C  DETAILED EXPERIMENTAL SETUP

### C.1  HARDWARE SPECIFICATION AND ENVIRONMENT

We run our experiments on a single machine with Dual Intel Xeon CPUs (E5-2698 v4 @ 2.2Ghz), one NVIDIA Tesla P100 GPU (16GB of HBM2 memory) and 512GB DDR4 memory. The code is written in Python 3.6.8 (where the sampling part is written with Cython 0.29.2). We use Tensorflow 1.12.0 on CUDA 9.2 with CUDNN 7.2.1 to train the model on GPU. Since the subgraphs are sampled independently, we run the sampler in parallel on 40 CPU cores.

---

**Algorithm 2** Graph sampling algorithms by `GraphSAINT`

---

**Input:** Training graph $\mathcal{G}\left(\mathcal{V}, \mathcal{E}\right)$; Sampling parameters: node budget $n$; edge budget $m$; number of roots $r$; random walk length $h$
**Output:** Sampled graph $\mathcal{G}_s\left(\mathcal{V}_s, \mathcal{E}_s\right)$

1: **function** NODE($\mathcal{G}$,n)                                         ▷ Node sampler
2:      $P\left(v\right) := \left\|\widetilde{\boldsymbol{A}}_{:,v}\right\|^2 / \sum_{v' \in \mathcal{V}} \left\|\widetilde{\boldsymbol{A}}_{:,v'}\right\|^2$
3:      $\mathcal{V}_s \leftarrow n$ nodes randomly sampled (with replacement) from $\mathcal{V}$ according to $P$
4:      $\mathcal{G}_s \leftarrow$ Node induced subgraph of $\mathcal{G}$ from $\mathcal{V}_s$
5: **end function**
6: **function** EDGE($\mathcal{G}$,m)                             ▷ Edge sampler (approximate version)
7:      $P\left((u,v)\right) := \left(\frac{1}{\deg(u)} + \frac{1}{\deg(v)}\right) / \sum_{(u',v') \in \mathcal{E}} \left(\frac{1}{\deg(u')} + \frac{1}{\deg(v')}\right)$
8:      $\mathcal{E}_s \leftarrow m$ edges randomly sampled (with replacement) from $\mathcal{E}$ according to $P$
9:      $\mathcal{V}_s \leftarrow$ Set of nodes that are end-points of edges in $\mathcal{E}_s$
10:      $\mathcal{G}_s \leftarrow$ Node induced subgraph of $\mathcal{G}$ from $\mathcal{V}_s$
11: **end function**
12: **function** RW($\mathcal{G}$,r,h)                                  ▷ Random walk sampler
13:      $\mathcal{V}_{\text{root}} \leftarrow r$ root nodes sampled uniformly at random (with replacement) from $\mathcal{V}$
14:      $\mathcal{V}_s \leftarrow \mathcal{V}_{\text{root}}$
15:      **for** $v \in \mathcal{V}_{\text{root}}$ **do**
16:          $u \leftarrow v$
17:          **for** $d = 1$ to $h$ **do**
18:              $u \leftarrow$ Node sampled uniformly at random from $u$'s neighbor
19:              $\mathcal{V}_s \leftarrow \mathcal{V}_s \cup \{u\}$
20:          **end for**
21:      **end for**
22:      $\mathcal{G}_s \leftarrow$ Node induced subgraph of $\mathcal{G}$ from $\mathcal{V}_s$
23: **end function**
24: **function** MRW($\mathcal{G}$,n,r)                        ▷ Multi-dimensional random walk sampler
25:      $\mathcal{V}_{\text{FS}} \leftarrow r$ root nodes sampled uniformly at random (with replacement) from $\mathcal{V}$
26:      $\mathcal{V}_s \leftarrow \mathcal{V}_{\text{FS}}$
27:      **for** $i = r + 1$ to $n$ **do**
28:          Select $u \in \mathcal{V}_{\text{FS}}$ with probability $\deg(u) / \sum_{v \in \mathcal{V}_{\text{FS}}} \deg(v)$
29:          $u' \leftarrow$ Node randomly sampled from $u$'s neighbor
30:          $\mathcal{V}_{\text{FS}} \leftarrow (\mathcal{V}_{\text{FS}} \setminus \{u\}) \cup \{u'\}$
31:          $\mathcal{V}_s \leftarrow \mathcal{V}_s \cup \{u\}$
32:      **end for**
33:      $\mathcal{G}_s \leftarrow$ Node induced subgraph of $\mathcal{G}$ from $\mathcal{V}_s$
34: **end function**

---

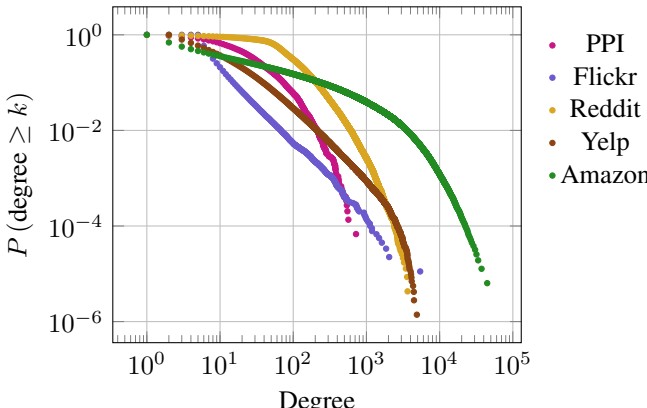

Figure 5: Degree Distribution

## C.2   ADDITIONAL DATASET DETAILS

Here we present the detailed procedures to prepare the Flickr, Yelp and Amazon datasets.

The Flickr dataset originates from NUS-wide[§]. The SNAP website[¶] collected Flickr data from four different sources including NUS-wide, and generated an un-directed graph. One node in the graph represents one image uploaded to Flickr. If two images share some common properties (e.g., same geographic location, same gallery, comments by the same user, etc.), there is an edge between the nodes of these two images. We use as the node features the 500-dimensional bag-of-word representation of the images provided by NUS-wide. For labels, we scan over the 81 tags of each image and manually merged them to 7 classes. Each image belongs to one of the 7 classes.

The Yelp dataset is prepared from the raw `json` data of businesses, users and reviews provided in the open challenge website[‖]. For nodes and edges, we scan the friend list of each user in the raw `json` file of users. If two users are friends, we create an edge between them. We then filter out all the reviews by each user and separate the reviews into words. Each review word is converted to a 300-dimensional vector using the Word2Vec model pre-trained on GoogleNews[**]. The word vectors of each node are added and normalized to serve as the node feature (i.e., $\boldsymbol{x}_v$). As for the node labels, we scan the raw `json` file of businesses, and use the categories of the businesses reviewed by a user $v$ as the multi-class label of $v$.

For the Amazon dataset, a node is a product on the Amazon website and an edge $(u, v)$ is created if products $u$ and $v$ are bought by the same customer. Each product contains text reviews (converted to 4-gram) from the buyer. We use SVD to reduce the dimensionality of the 4-gram representation to 200, and use the obtained vectors as the node feature. The labels represent the product categories (e.g., books, movies, shoes).

Figure 5 shows the degree distribution of the five graphs. A point $(k, p)$ in the plot means the probability of a node having degree at least $k$ is $p$.

## C.3   ADDITIONAL DETAILS IN EXPERIMENTAL CONFIGURATION

Table 4 summarizes the URLs to download the baseline codes.

The optimizer for `GraphSAINT` and all baselines is Adam (Kingma & Ba, 2014). For all baselines and datasets, we perform grid search on the hyperparameter space defined by:

- Hidden dimension: $\{128, 256, 512\}$

---

[§]`http://lms.comp.nus.edu.sg/research/NUS-WIDE.htm`
[¶]`https://snap.stanford.edu/data/web-flickr.html`
[‖]`https://www.yelp.com/dataset`
[**]`https://code.google.com/archive/p/word2vec/`

Table 4: URLs and commit number to run baseline codes

| Baseline | URL | Commit |
|---|---|---|
| Vanilla GCN | github.com/williamleif/GraphSAGE | a0fdef |
| GraphSAGE | github.com/williamleif/GraphSAGE | a0fdef |
| FastGCN | github.com/matenure/FastGCN | b8e6e6 |
| S-GCN | github.com/thu-ml/stochastic_gcn | da7b78 |
| AS-GCN | github.com/huangwb/AS-GCN | 5436ec |
| ClusterGCN | github.com/google-research/google-research/tree/master/cluster_gcn | 99021e |

Table 5: Training configuration of `GraphSAINT` for Table 2

| Sampler | Dataset | Training | | Sampling | | | |
|---|---|---|---|---|---|---|---|
| | | Learning rate | Dropout | Node budget | Edge budget | Roots | Walk length |
| Node | PPI | 0.01 | 0.0 | 6000 | — | — | — |
| | Flickr | 0.01 | 0.2 | 8000 | — | — | — |
| | Reddit | 0.01 | 0.1 | 8000 | — | — | — |
| | Yelp | 0.01 | 0.1 | 5000 | — | — | — |
| | Amazon | 0.01 | 0.1 | 4500 | — | — | — |
| Edge | PPI | 0.01 | 0.1 | — | 4000 | — | — |
| | Flickr | 0.01 | 0.2 | — | 6000 | — | — |
| | Reddit | 0.01 | 0.1 | — | 6000 | — | — |
| | Yelp | 0.01 | 0.1 | — | 2500 | — | — |
| | Amazon | 0.01 | 0.1 | — | 2000 | — | — |
| RW | PPI | 0.01 | 0.1 | — | — | 3000 | 2 |
| | Flickr | 0.01 | 0.2 | — | — | 6000 | 2 |
| | Reddit | 0.01 | 0.1 | — | — | 2000 | 4 |
| | Yelp | 0.01 | 0.1 | — | — | 1250 | 2 |
| | Amazon | 0.01 | 0.1 | — | — | 1500 | 2 |
| MRW | PPI | 0.01 | 0.1 | 8000 | — | 2500 | — |
| | Flickr | 0.01 | 0.2 | 12000 | — | 3000 | — |
| | Reddit | 0.01 | 0.1 | 8000 | — | 1000 | — |
| | Yelp | 0.01 | 0.1 | 2500 | — | 1000 | — |
| | Amazon | 0.01 | 0.1 | 4500 | — | 1500 | — |

- Dropout: $\{0.0, 0.1, 0.2, 0.3\}$

- Learning rate: $\{0.1, 0.01, 0.001, 0.0001\}$

The hidden dimensions used for Table 2, Figure 2, Figure 3 and Figure 4 are: 512 for PPI, 256 for Flickr, 128 for Reddit, 512 for Yelp and 512 for Amazon.

All methods terminate after a fixed number of epochs based on convergence. We save the model producing the highest validation set F1-micro score, and reload it to evaluate the test set accuracy.

For vanilla GCN and AS-GCN, we set the batch size to their default value 512. For GraphSAGE, we use the mean aggregator with the default batch size 512. For S-GCN, we set the flag `-cv -cvd` (which stand for "control variate" and "control variate dropout") with pre-computation of the first layer aggregation. According to the paper (Chen et al., 2018a), such pre-computation significantly reduces training time without affecting accuracy. For S-GCN, we use the default batch size 1000, and for FastGCN, we use the default value 400. For ClusterGCN, its batch size is determined by two parameters: the cluster size and the number of clusters per batch. We sweep the cluster size from 500 to 10000 with step 500, and the number of clusters per batch from $\{1, 10, 20, 40\}$ to determine the optimal configuration for each dataset / architecture. Considering that for ClusterGCN, the cluster structure may be sensitive to the cluster size, and for FastGCN, the minibatch connectivity may increase with the sample size, we present additional experimental results to reveal the relation between accuracy and batch size in Appendix D.3.

Table 6: Training configuration of `GraphSAINT` for Table 3

| Arch. | Sampler | Dataset | Training | | Sampling | | | |
| --- | --- | --- | --- | --- | --- | --- | --- | --- |
| | | | Learning rate | Dropout | Node budget | Edge budget | Roots | Walk length |
| $2 \times 512$ | MRW | PPI (large) | 0.01 | 0.1 | 1500 | — | 300 | — |
| $5 \times 2048$ | RW | PPI (large) | 0.01 | 0.1 | — | — | 3000 | 2 |
| $2 \times 128$ | Edge | Reddit | 0.01 | 0.1 | — | 6000 | — | — |
| $4 \times 128$ | Edge | Reddit | 0.01 | 0.2 | — | 11000 | — | — |

Table 7: Training configuration of `GraphSAINT` for Figure 4 (Reddit)

| | 2-layer GAT-SAINT | 4-layer GAT-SAINT | 2-layer JK-SAINT | 4-layer JK-SAINT |
| --- | --- | --- | --- | --- |
| Hidden dimension | 128 | 128 | 128 | 128 |
| Attention $K$ | 8 | 8 | — | — |
| Aggregation $\bigoplus$ | — | — | Concat. | Concat. |
| Sampler | RW (root: 3000; length: 2) | RW (root: 2000; length: 4) | Edge (budget: 6000) | Edge (budget: 11000) |
| Learning rate | 0.01 | 0.01 | 0.01 | 0.01 |
| Dropout | 0.2 | 0.2 | 0.1 | 0.2 |

Configuration of `GraphSAINT` to reproduce Table 2 results is shown in Table 5. Configuration of `GraphSAINT` to reproduce Table 3 results is shown in Table 6.

Below we describe the configuration for Figure 4.

The major difference between a normal GCN and a JK-net (Xu et al., 2018) is that JK-net has an additional final layer that aggregates all the output hidden features of graph convolutional layers 1 to $L$. Mathematically, the additional aggregation layer outputs the final embedding $\boldsymbol{x}_{\text{JK}}$ as follows:

$$\boldsymbol{x}_{\text{JK}} = \sigma \left( \boldsymbol{W}_{\text{JK}}^{\mathsf{T}} \cdot \bigoplus_{\ell=1}^{L} \boldsymbol{x}_v^{(\ell)} \right) \tag{7}$$

where based on Xu et al. (2018), $\bigoplus$ is the vector aggregation operator: max-pooling, concatenation or LSTM (Hochreiter & Schmidhuber, 1997) based aggregation.

The graph attention of GAT (Veličković et al., 2017) calculates the edge weights for neighbor aggregation by an additional neural network. With multi-head ($K$) attention, the layer-$(\ell-1)$ features propagate to layer-$(\ell)$ as follows:

$$\boldsymbol{x}_v^{(\ell)} = \left\| \begin{array}{c} K \\ k=1 \end{array} \sigma \left( \sum_{u \in \text{neighbor}(v)} \alpha_{u,v}^k \boldsymbol{W}^k \boldsymbol{x}_v^{(\ell-1)} \right) \right. \tag{8}$$

where $\|$ is the vector concatenation operation, and the coefficient $\alpha$ is calculated with the attention weights $\boldsymbol{a}^k$ by:

$$\alpha_{u,v}^k = \text{LeakyReLU} \left( \left( \boldsymbol{a}^k \right)^{\mathsf{T}} \left[ \boldsymbol{W}^k \boldsymbol{x}_u \| \boldsymbol{W}^k \boldsymbol{x}_v \right] \right) \tag{9}$$

Note that the $\alpha$ calculation is slightly different from the original equation in Veličković et al. (2017). Namely, GAT-SAINT does not normalize $\alpha$ by softmax across all neighbors of $v$. We make such modification since under the minibatch setting, node $v$ does not see all its neighbors in the training graph. The removal of softmax is also seen in the attention design of Huang et al. (2018). Note that during the minibatch training, GAT-SAINT further applies another edge coefficient on top of attention for aggregator normalization.

Table 7 shows the configuration of the GAT-SAINT and JK-SAINT curves in Figure 4.

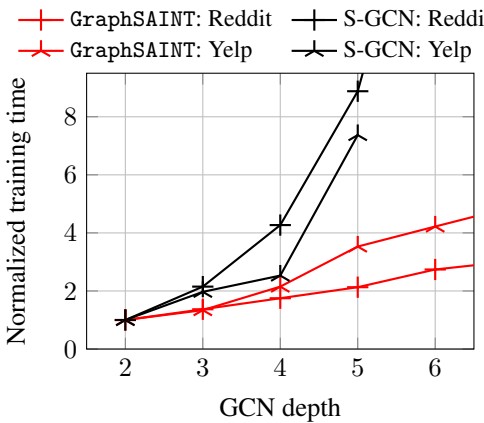
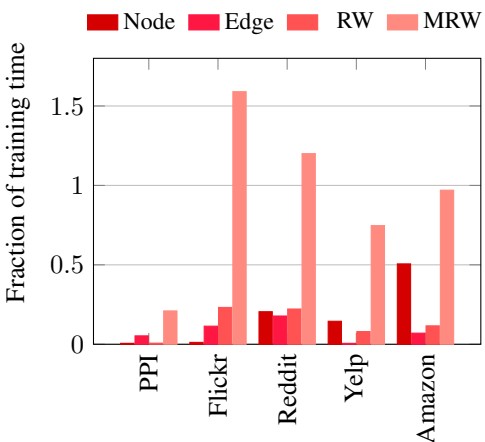

Figure 6: Comparison of training efficiency  Figure 7: Fraction of training time on sampling

# D  ADDITIONAL EXPERIMENTS

## D.1  TRAINING EFFICIENCY ON DEEP MODELS

We evaluate the training efficiency for deeper GCNs. We only compare with S-GCN, since implementations for other layer sampling based methods have not yet supported arbitrary model depth. The batch size and hidden dimension are the same as Table 2. On the two large graphs (Reddit and Yelp), we increase the number of layers and measure the average time per minibatch execution. In Figure 6, training cost of GraphSAINT is approximately linear with GCN depth. Training cost of S-GCN grows dramatically when increasing the depth. This reflects the "neighbor explosion" phenomenon (even though the expansion factor of S-GCN is just 2). On Yelp, S-GCN gives "out-of-memory" error for models beyond 5 layers.

## D.2  COST OF SAMPLING AND PRE-PROCESSING

**Cost of graph samplers of GraphSAINT**  Graph sampling introduces little training overhead. Let $t_s$ be the average time to sample one subgraph on a multi-core machine. Let $t_t$ be the average time to perform the forward and backward propagation on one minibatch on GPU. Figure 7 shows the ratio $t_s/t_t$ for various datasets. The parameters of the samplers are the same as Table 2. For Node, Edge and RW samplers, we observe that time to sample one subgraph is in most cases less than 25% of the training time. The MRW sampler is more expensive to execute. Regarding the complete pre-processing procedure, we repeatedly run the sampler for $N = 50 \cdot |\mathcal{V}| / \overline{|\mathcal{V}_s|}$ times before training, to estimate the node and edge probability as discussed in Section 3.2 (where $\overline{|\mathcal{V}_s|}$ is the average subgraph size). These sampled subgraphs are reused as training minibatches. Thus, if training runs for more than $N$ iterations, the pre-processing is nearly zero-cost. Under the setting of Table 2, pre-processing on PPI and Yelp and Amazon does not incur any overhead in training time. Pre-processing on Flickr and Reddit (with RW sampler) takes less than 40% and 15% of their corresponding total training time.

**Cost of layers sampler of AS-GCN**  AS-GCN uses an additional neural network to estimate the conditional sampling probability for the previous layer. For a node $v$ already sampled in layer $\ell$, features of layer-$(\ell-1)$ corresponding to all $v$'s neighbors need to be fed to the sampling neural network to obtain the node probability. For sake of analysis, assume the sampling network is a single layer MLP, whose weight $\boldsymbol{W}_{\text{MLP}}$ has the same shape as the GCN weights $\boldsymbol{W}^{(\ell)}$. Then we can show, for a $L$-layer GCN on a degree-$d$ graph, per epoch training complexity of AS-GCN is approximately $\gamma = (d \cdot L) / \sum_{\ell=0}^{L-1} d^\ell$ times that of vanilla GCN. For $L = 2$, we have $\gamma \approx 2$. This explains the observation that AS-GCN is slower than vanilla GCN in Figure 2. Additional, Table 8 shows the training time breakdown for AS-GCN. Clearly, its sampler is much more expensive than the graph sampler of GraphSAINT.

Table 8: Per epoch training time breakdown for AS-GCN

| Dataset | Sampling time (sec) | Forward / Backward propagation time (sec) |
|---------|---------------------|-------------------------------------------|
| PPI | 1.1 | 0.2 |
| Flickr | 5.3 | 1.1 |
| Reddit | 20.7 | 3.5 |

**Cost of clustering of ClusterGCN**  ClusterGCN uses the highly optimized METIS software[††] to perform clustering. Table 9 summarizes the time to obtain the clusters for the five graphs. On the large and dense Amazon graph, the cost of clustering increase dramatically. The pre-processing time of ClusterGCN on Amazon is more than $4\times$ of the total training time. On the other hand, the sampling cost of GraphSAINT does not increase significantly for large graphs (see Figure 7).

Table 9: Clustering time of ClusterGCN

|  | PPI | Flickr | Reddit | Yelp | Amazon |
|--|-----|--------|--------|------|--------|
| Time (sec) | 2.2 | 11.6 | 40.0 | 106.7 | 2254.2 |

Taking into account the pre-processing time, sampling time and training time altogether, we summarize the total convergence time of GraphSAINT and ClusterGCN in Table 10 (corresponding to Table 2 configuration). On graphs that are large and dense (e.g., Amazon), GraphSAINT achieves significantly faster convergence. Note that both the sampling of GraphSAINT and clustering of ClusterGCN can be performed offline.

Table 10: Comparison of total convergence time (pre-processing + sampling + training, unit: second)

|  | PPI | Flickr | Reddit | Yelp | Amazon |
|--|-----|--------|--------|------|--------|
| GraphSAINT-Edge | 91.0 | 7.0 | 16.6 | 273.9 | 401.0 |
| GraphSAINT-RW | 103.6 | 7.5 | 17.2 | 310.1 | 425.6 |
| ClusterGCN | 163.2 | 12.9 | 55.3 | 256.0 | 2804.8 |

### D.3 EFFECT OF BATCH SIZE

Table 11 shows the change of test set accuracy with batch sizes. For each row of Table 11, we fix the batch size, tune the other hyperparameters according to Appendix C.3, and report the highest test set accuracy achieved. For GraphSAGE, S-GCN and AS-GCN, their default batch sizes (512,1000 and 512, respectively) lead to the highest accuracy on all datasets. For FastGCN, increasing the default batch size (from 400 to 4000) leads to noticeable accuracy improvement. For ClusterGCN, different datasets correspond to different optimal batch sizes. Note that the accuracy in Section 5.1 is already tuned by identifying the optimal batch size on a per graph basis.

For FastGCN, intuitively, increasing batch size may help with accuracy improvement since the minibatches may become better connected. Such intuition is verified by the rows of 400 and 2000. However, increasing the batch size from 2000 to 4000 does not further improve accuracy significantly. For ClusterGCN, the optimal batch size depends on the cluster structure of the training graph. For PPI, small batches are better, while for Amazon, batch size does not have significant impact on accuracy. For GraphSAGE, overly large batches may have negative impact on accuracy due to neighbor explosion. Approximately, GraphSAGE expand $10\times$ more neighbors per layer. For a 2-layer GCN, a size $2 \times 10^3$ minibatch would then require the support of $2 \times 10^5$ nodes from the

---

[††]http://glaros.dtc.umn.edu/gkhome/metis/metis/download
[*]Default batch size
[¶]The training does not converge.
[‡]The codes throw runtime error on the large datasets (Yelp or Amazon).

Table 11: Test set F1-micro for the baselines under various batch sizes

| Method | Batch size | PPI | Flickr | Reddit | Yelp | Amazon |
|---|---|---|---|---|---|---|
| GraphSAGE | 256 | 0.600 | 0.474 | 0.934 | 0.563 | 0.428 |
| | 512* | 0.637 | 0.501 | 0.953 | 0.634 | 0.758 |
| | 1024 | 0.610 | 0.482 | 0.935 | 0.632 | 0.705 |
| | 2048 | 0.625 | 0.374 | 0.936 | 0.563 | 0.447 |
| FastGCN | 400* | 0.513 | 0.504 | 0.924 | 0.265 | 0.174 |
| | 2000 | 0.561 | 0.506 | 0.934 | 0.255 | 0.196 |
| | 4000 | 0.564 | 0.507 | 0.934 | 0.260 | 0.195 |
| S-GCN | 500 | 0.519 | 0.462 | —¶ | —¶ | —‡ |
| | 1000* | 0.963 | 0.482 | 0.964 | 0.640 | —‡ |
| | 2000 | 0.646 | 0.482 | 0.949 | 0.614 | —‡ |
| | 4000 | 0.804 | 0.482 | 0.949 | 0.594 | —‡ |
| | 8000 | 0.694 | 0.481 | 0.950 | 0.613 | —‡ |
| AS-GCN | 256 | 0.682 | 0.504 | 0.950 | —‡ | —‡ |
| | 512* | 0.687 | 0.504 | 0.958 | —‡ | —‡ |
| | 1024 | 0.687 | 0.502 | 0.951 | —‡ | —‡ |
| | 2048 | 0.670 | 0.502 | 0.952 | —‡ | —‡ |
| ClusterGCN | 500 | 0.875 | 0.481 | 0.942 | 0.604 | 0.752 |
| | 1000 | 0.831 | 0.478 | 0.947 | 0.602 | 0.756 |
| | 1500 | 0.865 | 0.480 | 0.954 | 0.602 | 0.752 |
| | 2000 | 0.828 | 0.469 | 0.954 | 0.609 | 0.759 |
| | 2500 | 0.849 | 0.476 | 0.954 | 0.598 | 0.745 |
| | 3000 | 0.840 | 0.473 | 0.954 | 0.607 | 0.754 |
| | 3500 | 0.846 | 0.473 | 0.952 | 0.602 | 0.754 |
| | 4000 | 0.853 | 0.472 | 0.949 | 0.605 | 0.756 |

input layer. Note that the full training graph size of Reddit is just around $1.5 \times 10^5$. Thus, no matter which nodes are sampled in the output layer, GraphSAGE would almost always propagate features within the full training graph for initial layers. We suspect this would lead to difficulties in learning. For S-GCN, with batch size of 500, it fails to learn properly on Reddit and Yelp. The accuracy fluctuates in a region of very low value, even after appropriate hyperparameter tuning. For AS-GCN, its accuracy is not sensitive to the batch size, since AS-GCN addresses neighbor explosion and also ensures good inter-layer connectivity within the minibatch.

