# OpenReview forum: "GraphSAINT: Graph Sampling Based Inductive Learning Method"
_ICLR.cc/2020/Conference — Accept (Poster)_

### Official Review · AnonReviewer1 · 2019-10-21
**Official Blind Review #1**

**Rating:** 6

**Review:**

This paper proposed a new sampling method to train GCN in the mini-batch manner. In particular, unlike existing methods which samples the mini-batch in the node-wise way, GraphSAINT proposed to sample a mini-batch in the graph-wise way. As a result, GraphSAINT uses the same graph across different GCN layers, while most existing methods use different graphs across different GCN layers.  In addition, the authors show that this sampling method is unbiased. Extensive experimental results have shown improvement over existing methods. Overall, this idea is interesting and well presented.

Pros:
1. A new sampling method for the stochastic training of GCN. Have good performance.
2. Extensive experiments to verify the performance of the proposed method.
3. The theoretical analysis looks sound.

Cons:
1. GraphSAGE and FastGCN use different graphs across different GCN layers, while ClusterGCN and GraphSAINT use the same graph across different GCN layers. To make a fair comparison, it is necessary to have the same batch size for different methods. How do you deal with this issue in your experiment?
2. For ClusterGCN, the clustering procedure is done before the training. So, it needs much less computational overhead for sampling in the training course. However, GraphSAINT needs to do the heavy sampling online. Thus, it may consume more time than ClusterGCN for large graphs. It's better to show the running time of these two methods.


**Experience Assessment:**

I have published in this field for several years.

**Review Assessment: Checking Correctness Of Derivations And Theory:**

I assessed the sensibility of the derivations and theory.

**Review Assessment: Checking Correctness Of Experiments:**

I assessed the sensibility of the experiments.

**Review Assessment: Thoroughness In Paper Reading:**

I read the paper thoroughly.

---

> ### Author Response · Authors · 2019-11-08
> **Response to Review #1**
>
> We appreciate the valuable feedback from the reviewer! In the following, we would like to clarify the two "Cons":
>
>
> 1. Regarding batch size (defined by all the baselines as the number of node samples in the output layer):
>
> As noted in the review, the four methods (GraphSAGE, FastGCN, S-GCN, AS-GCN) use samples of different sizes across different layers, even when the batch size is set to be the same. This leads to the intuition that the optimal batch size (w.r.t. accuracy) should be different for different methods. In the experiments, we have treated batch size as a hyperparameter dependent on the sampling method as well as the training graph topology.
>
> By experiments on varying the batch sizes, we observe: for GraphSAGE, S-GCN and AS-GCN, their default batch sizes (512,1000 and 512, respectively) lead to the highest accuracy on all datasets. For FastGCN, increasing the default batch size (from 400 to 4000) leads to noticeable accuracy improvement. For ClusterGCN, different datasets correspond to different optimal batch sizes, and the accuracy in Section 5.1 is already tuned by identifying the optimal batch size on a per graph basis. See also Table 11 (or Table 10 in the original submission) of Appendix D.3 for experiment details.
>
>
> 2. Regarding sampling overhead:
>
> As discussed in Appendix D.2, the two best samplers of GraphSAINT, "Edge" and "RW", are very light-weight. In addition, similar to ClusterGCN, our sampling can also be done offline since the sampler does not require node features. To be more specific, time to construct one subgraph by "Edge" or "RW" is always less than 25% of the time to perform one gradient update. On the other hand, as shown in Table 9, time to identify clusters by ClusterGCN can be much longer than its total training time if the training graph is large and dense. For example, clustering time on Amazon is over 5x the total training time of ClusterGCN.
>
> Taking into account the pre-processing time, sampling time and training time altogether, we summarize the total convergence time (in seconds) of GraphSAINT and ClusterGCN in the following (corresponding to Table 2 configuration):
> ---------------------------------------------------------------------------------------
>                                       PPI       Flickr   Reddit   Yelp       Amazon
> ---------------------------------------------------------------------------------------
> GraphSAINT-Edge     91.0       7.0      16.6       273.9      401.0
> GraphSAINT-RW        103.6     7.5      17.2       310.1      425.6
> ClusterGCN                163.2     12.9     55.3      256.0      2804.8
> ----------------------------------------------------------------------------------------

---

> > ### Author Response · Authors · 2019-11-15
> > **New Revision of the Paper Uploaded**
> >
> > We have uploaded a new version of the paper which includes new experimental results.
> >
> > In Appendix D.3, we have included the test set accuracy of baselines under various batch sizes in Table 11. The results support the "point 1" in our previous response.
> >
> > In Appendix D.2, we have included an additional table comparing the total convergence time of GraphSAINT and ClusterGCN after considering the pre-processing cost and sampling cost. The results are in line with the "point 2" in our previous response.

---

### Official Review · AnonReviewer3 · 2019-10-23
**Official Blind Review #3**

**Rating:** 6

**Review:**

This paper proposes a training method for graph convolution networks on large graphs. The idea is to train a full GCN on partial samples of the graph. The graph samples are computed based on the graph connectivity, and the authors propose methods for reducing the bias and variance in the training procedure.

The idea is elegant and intuitive, and the fact that the approach can work with various graph sampling methods adds to its generality. The paper is well-written and the fact that code is published is valuable.

The results on bias and variance are under the assumption that each layer independently learns an embedding. This would be clearer if added explicitly in the theorem statements (and not as part of the main text). It would be interesting to discuss how realistic this assumption is, and how large the actual bias is. Perhaps this can be measured empirically?
Nevertheless, the empirical result indeed support the claim that this simplifying assumption is enough to derive useful learning rules.

Overall, I believe this is a solid contribution, and I can foresee future extensions that improve the results with more complex graph sampling methods.

Question to the authors: I did not understand the second equality in Eq. 3. Could there be a typo?


**Experience Assessment:**

I do not know much about this area.

**Review Assessment: Checking Correctness Of Derivations And Theory:**

I assessed the sensibility of the derivations and theory.

**Review Assessment: Checking Correctness Of Experiments:**

I assessed the sensibility of the experiments.

**Review Assessment: Thoroughness In Paper Reading:**

I made a quick assessment of this paper.

---

> ### Author Response · Authors · 2019-11-08
> **Response to Review #3**
>
> Thanks a lot for your valuable feedback. We will state our assumption on the theorem more explicitly in our next revision.
>
> Answer to the question:
>
> Yes, there is a typo in Equation 3. Thanks for pointing this out! The correct expression should be $\mathbb{E}(L_\text{batch})=\frac{1}{|\mathbb{G}|} \sum\limits_{\mathcal{G}_s \in \mathbb{G}}\sum\limits_{v\in \mathcal{V}_s} \frac{L_v}{\lambda_v}=\frac{1}{|\mathcal{V}|}\sum\limits_{v\in\mathcal{V}} L_v$.

---

> > ### Author Response · Authors · 2019-11-15
> > **New Revision of the Paper Uploaded**
> >
> > We have uploaded a new version of the paper after integrating your constructive suggestions.
> >
> > Regarding clarifying the theorem statement:
> >
> > We have updated the text in Section 3.2 as well as the statement of Proposition 3.1. Note that for given ${x}_u^{(\ell)}$, Proposition 3.1 itself considers a single layer $\ell+1$ and does not rely on the assumption that "each layer learns embeddings independently". On the other hand, as noted by the reviewer, such assumption is required when using the proposition to normalize the multi-layer GCN built by GraphSAINT. Therefore, in the updated paper, we clarify such assumption right before and after the statement of Proposition 3.1.

---

### Official Review · AnonReviewer2 · 2019-11-03
**Official Blind Review #2**

**Rating:** 6

**Review:**

Scaling GCNs to large graphs is important for real applications. Instead of sampling the nodes or edges across GCN layers,  this paper proposes to sample the training graph to improve training efficiency and accuracy. It is a smart idea to construct a complete GCN from the sampled subgraphs.  Convincing experiments can verify the effectiveness of the proposed method.  It is a good work.

Question:
1. How can the authors guarantee that subgraphs are properly sampled? Are there any  theoretical guarantee?


**Experience Assessment:**

I have read many papers in this area.

**Review Assessment: Checking Correctness Of Derivations And Theory:**

I assessed the sensibility of the derivations and theory.

**Review Assessment: Checking Correctness Of Experiments:**

I carefully checked the experiments.

**Review Assessment: Thoroughness In Paper Reading:**

I made a quick assessment of this paper.

---

> ### Author Response · Authors · 2019-11-13
> **Response to Review #2**
>
> Thank you for your valuable feedback. We agree that properly sampled subgraphs are critical to high accuracy, and sampling parameters need to be carefully chosen. In fact, we design samplers based on the theoretical analysis on bias and variance of the minibatch estimator.
>
> To eliminate bias introduced by graph sampling, we derive the normalization on feature aggregator and minibatch loss. Note that such normalization ensures unbiasedness for an arbitrary graph sampler (Proposition 3.1) . To minimize the variance of the minibatch estimator, we derive the optimal sampling parameter for "Edge" sampler (Theorem 3.2). We further extend the proposed edge sampler to random walk samplers and determine the corresponding sampling parameters, based on insights into the GCN architecture.
>
> Note that our samplers derived from the theoretical analysis also satisfies the intuitive requirement for a "proper" sampler -- that nodes influential to each other should have high probability to be sampled together. Please see Section 3.3 for a detailed discussion.
>
> Our experiments show that the choices of normalization and graph samplers based on Proposition 3.1 and Theorem 3.2 do lead to improved accuracy. As shown in Table 2, accuracy results of GraphSAINT are indeed state-of-the-art.

---

### Public Comment · ~Weilin_Cong1 · 2019-10-17
**GraphSaint is not unbiased**

Thanks for your paper, it is indeed a very interesting idea. However, I cannot agree with all the claims you made.

For example in Proposition 3.1 you claim as $\xi_v^{(l+1)}$ is an unbiased estimator of the aggregation of $v$ in full GCN if $\alpha_{u,v} = p_{u,v}/p_v$, i.e., $\mathbb{E}(\xi_v^{(l+1)}) = \sum_{u\in V} A_{v,u}x_u^{(l)}$.

However, I think $\xi_v^{(l+1)}$ is unbiased only condition on the last layer feature $x_u^{(l)}$, i.e.,  $\mathbb{E}(\xi_v^{(l+1)} | x_u^{(l)}) = \sum_{u\in V} A_{v,u}x_u^{(l)} $ due to the non-linear activations. You cannot ignore the non-linear activation in your proof since $\mathbb{E}(\sigma(x)) \neq \sigma(\mathbb{E}(x))$ if $\sigma()$ is an non-linear activation.

Please clarify if possible. Thanks.

---

> ### Author Response · Authors · 2019-10-18
> **GraphSAINT is unbiased under our assumption**
>
> Thanks for your interest in our paper. This is a valid concern.
>
> As stated at the beginning of Section 3.2, “analysis of the complete multi-layer GCN is difficult due to non-linear activations. Thus, we analyze the embedding of each layer independently.” In other words, to derive Proposition 3.1, we followed the same assumption as AS-GCN and FastGCN, that each layer independently learns an embedding. Thus, the condition on the previous layer can be removed. This assumption also motivates the proposed edge sampler.
>
> Alternatively, in Section 3.4, we have also performed analysis by assuming no non-linear activations. Then we can collapse L layers of A into an equivalent 1 layer of A^L. Analysis based on this assumption leads to the proposed random walk sampler.
>
> In practice, it is possible that neither of the above two assumptions are exactly true, but they provide an approach to normalize loss and choose samplers and their parameters. Our experiments show that the choices of normalization and samplers based on these assumptions do lead to improved accuracy.

---

### Decision · Program_Chairs · 2019-12-19

**Decision:**

Accept (Poster)

**Comment:**

All three reviewers advocated acceptance. The AC agrees, feeling the paper is interesting.